# Effect of Milk Fat Globule Membrane- and Protein-Containing Snack Product on Physical Performance of Older Women—A Randomized Controlled Trial

**DOI:** 10.3390/nu15132922

**Published:** 2023-06-27

**Authors:** Satu K. Jyväkorpi, Riikka T. Niskanen, Marianna Markkanen, Karoliina Salminen, Timo Sibakov, Kaity-Marin Lehtonen, Susanna Kunvik, Kaisu H. Pitkala, Anu M. Turpeinen, Merja H. Suominen

**Affiliations:** 1Unit of Primary Health Care, Department of General Practice and Primary Health Care, Faculty of Medicine, Helsinki University Central Hospital, University of Helsinki, 00014 Helsinki, Finland; riikka.niskanen@helsinki.fi (R.T.N.); krln.salminen@gmail.com (K.S.); kaisu.pitkala@helsinki.fi (K.H.P.); merja.suominen@gery.fi (M.H.S.); 2Department of Food and Nutrition, University of Helsinki, 00014 Helsinki, Finland; marianna.markkanen@gery.fi; 3Society for Gerontological Nutrition in Finland, 00700 Helsinki, Finland; 4Valio Ltd., R&D, 00370 Helsinki, Finland; timo.sibakov@valio.fi (T.S.); kaity-marin.lehtonen@valio.fi (K.-M.L.); anu.turpeinen@valio.fi (A.M.T.); 5Faculty of Health and Welfare, Satakunta University of Applied Sciences, 28101 Pori, Finland; susanna.kunvik@samk.fi

**Keywords:** milk fat globule membrane, older women, protein, 5-time-chair-stand test, balance, SPPB

## Abstract

Introduction: Sarcopenia is common in people 70+ years of age, and its prevalence increases with further aging. Insufficient energy and protein intake accelerates muscle loss, whereas sufficient protein intake and milk fat globule membrane (MFGM) may suppress age-associated deterioration of muscle mass and strength. Our objective was to test whether a snack product high in MFGM and protein would improve physical performance in older women. Methods: In this 12-week randomized controlled trial, women ≥ 70 years, with protein intake < 1.2 g/body weight (BW) kg/day (d), were randomized into intervention (n = 51) and control (n = 50) groups. The intervention group received a daily snack product containing ≥ 23 g of milk protein and 3.6–3.9 g of MFGM. Both groups were advised to perform a five-movement exercise routine. The primary outcome was the change in the five-time-sit-to-stand test between the groups. Secondary outcomes included changes in physical performance, cognition, hand grip strength, and health-related quality of life. Results: The change in the five-time-sit-to-stand test did not differ between the intervention and the control groups. The change in the total Short Physical Performance Battery score differed significantly, favoring the intervention group (*p* = 0.020), and the balance test showed the largest difference. Protein intake increased significantly in the intervention group (+14 g) compared to the control group (+2 g). No other significant changes were observed. Conclusions: Our results indicate that the combination of MFGM and protein may improve the physical performance-related balance of older women.

## 1. Introduction

Good physical performance is central to active and healthy aging. Physical performance relates not only to the musculoskeletal system, but also to other aspects of daily living. Sarcopenia can be defined as low muscle strength, mass, and quality according to the European Working Group on Sarcopenia in Older People 2 group [1]. In addition to the previous characteristics, severe sarcopenia includes poor physical performance. The prevalence of sarcopenia increases with aging, especially in people > 70 years. Loss of muscle mass is associated with malnutrition, low protein intake, low physical activity, inflammation, chronic disease, unintentional weight loss, and low body mass index [1,2,3]. Both sarcopenia and frailty increase the risk of mobility disability, falls, poor physical function, decline in quality of life, institutionalization, and premature death [1,2,3,4]. As the world’s population is aging, it is of utmost importance to find feasible interventions to counteract this harmful cycle leading to these poor outcomes in older people.

Milk fat globule membrane (MFGM) is a complex structure composed primarily of lipids and proteins that surround fat globules in mammal milk. It is a source of multiple bioactive compounds, including phospholipids, glycolipids, and glycoproteins, that have important functional roles within the brain and gut [5]. MFGM is found to be antimicrobial, anti-inflammatory, anti-cholesterolemic, and safe to use as a supplement [6,7,8]. Supplementation of infant formula with MFGM has enhanced cognitive development in infants compared to standard infant formula [9,10], whereas effects of MFGM on cognitive functions in older individuals have not been reported.

Previous studies on physical function and performance have found that MFGM supplementation may stimulate neuromuscular junction development, which is a critical structure of motor units involved in physical movement [11]. Therefore, an increase in active motor units may lead to improvement in physical function and performance. Interventions including MFGM supplements combined with exercise have in fact improved ambulatory activities, leg muscle mass, and muscle fiber velocity in older adults [7]. MFGM supplementation given to middle-aged adults as part of a regular exercise regime enhanced physical agility by promoting fast-type motor unit maintenance compared to a control group [12]. Similarly, MFGM supplementation combined with twice-weekly physical activity improved frail older (>75 years) women’s frailty status compared with exercise and placebo supplementation as well as MFGM supplementation alone [13]. 

So far, most of the research on MFGM supplementation on physical performance has been published from Japan [12,13,14,15]. There are no data on the effects of MFGM supplementation on other ethnic groups. Moreover, many older people consume less protein than recommended [16]. Hydrolyzed milk protein is absorbed more rapidly than intact protein and increases postprandial amino acid availability; thus, it may augment postprandial muscle protein synthetic response in older people [17]. Therefore, we decided to offer our study participants a snack product high in both MFGM and hydrolyzed protein. Thus, the purpose of this study was to test whether a snack product high both in MFGM and hydrolyzed protein would improve the physical performance of older women with protein intake below the recommended level at baseline.

## 2. Methods

In 2021, 101 older, home-dwelling women (≥70 years) were recruited to the study mainly via Facebook and a newspaper ad in the major Finnish newspaper *Helsingin Sanomat*. Sarcopenia was screened with the SARC-F questionnaire [18]. The subjects were excluded from the study if they scored 0 points on the SARC-F questionnaire [18], had a diagnosed memory disorder or poor cognition Mini-Mental State Examination (MMSE) score < 24 [19], or were unable to move independently. All the inclusion and exclusion criteria are presented in more detail in Table 1. Because our study took place during an ongoing pandemic, we only accepted participants who had been vaccinated against COVID-19 due to safety reasons, although this was not an exclusion criterion in our original study plan. Oral and written informed consent was obtained from the participants after they were fully informed about the methods of the study. The ethics committee of the Department of Medicine at Helsinki University Hospital and City of Helsinki approved the study protocol (No. HUS/3022/2020). The trial was registered by the ACTRN, registration No. ACTRN12621001630808.

### 2.1. Study Protocol

The participants were randomly assigned to the intervention or control group. The intervention group received a daily snack product containing MFGM and protein for 12 weeks. In addition, both groups (intervention and control) were taught and advised on a simple five-movement exercise routine, which they were encouraged to independently perform every day during the study. To monitor their compliance with the advice, the participants filled out a daily track sheet, where they reported whether they had consumed a snack product and/or performed the exercise.

### 2.2. Advice

A trained dietician gave advice by phone to the participants in the intervention group about the consumption of the snack products provided by the study. In case their BMI was >23 kg/m^2^, the dietician advised the participants to replace a usually consumed snack (e.g., bread or pastry) in their daily diets with the study product. If their BMI was ≤23 kg/m^2^, the participants were encouraged to consume the snack product in addition to their usual diet. The participants were also advised to consume the snack within an hour following the exercise routine.

### 2.3. The Snack Product

We offered two types of products to the participants in the intervention group: a chocolate milkshake (a serving of 250 mL) and protein powder (a serving of 30 g). Lactose-free, protein-hydrolyzed buttermilk powder and chocolate-flavored milkshakes were produced from ultrafiltered lactose-free buttermilk concentrate. In ultrafiltration, some of the monosaccharides were removed, and proteins and residual fats were concentrated. After ultrafiltration, part of the proteins was hydrolyzed by enzymatic hydrolysis. The lactose content of the final protein concentrate was <0.01%. Protein hydrolysis was undertaken according to patent EP 2632277B1 [20] and as described previously [21]. The serving size of both products contained the same amount of protein (23 g) and a similar amount of MFGM (powder 3.9 g and milkshake 3.6 g). A more precise nutritional content of the snack products is presented in Table 2. The snack products were manufactured by the study sponsor Valio Ltd. 

### 2.4. Exercise Routine

At the baseline visit, each participant in both groups was taught a short, five-movement exercise routine which they were encouraged to perform daily. The routine was planned to maintain muscle mass and strength of the lower extremities. It was designed by a non-governmental organization, Age Institute, which is an expert organization specialized in enhancing healthy aging (https://www.ikainstituutti.fi/in-english/ (accessed on 10 January 2023). The exercise routine is available at the following link: https://www.ikainstituutti.fi/content/uploads/2021/02/KAVELY_KEVYEMMAKSI_ENG_saav0.pdf (accessed on 1 September 2021).

### 2.5. Measurements

The participants were followed up for 12 weeks. The primary outcome of the study was the difference in changes in the five-time-sit-to-stand test between the intervention and the control groups. The five-time-sit-to-stand test was measured during baseline and end visits at 12 weeks. The test is an indicator of muscular endurance and agility and is a good predictor of mobility disability and falls in home-dwelling older people [22]. During the test, the participants stood up and sat down five times with their arms folded in front of their chest as quickly as possible on a firm chair [23]. The time required to complete five cycles was measured. The test is also part of the Short Physical Performance Battery (SPPB) test, which also includes walking speed and balance tests [24]. The SPPB test in its entirety was included in the measurements. Grip strength was measured with a dynamometer (Saehan DHD-1 Digital hand dynamometer) “using a standard protocol”.

Cognition was measured using Trail Making Test (TMT) A and B, which test attention and psychomotor speed [25]. TMT B also requires good executive function (Reitan 1958). Health-related quality of life was assessed using the RAND-36 test, which is also validated in the Finnish population [26,27].

Energy and protein intakes were measured using a 3-day food diary before the baseline visit to verify whether potential participants met the inclusion criteria on protein intake (protein intake < 1.2 g/kg body weight (BW)/d). In addition, 3-day food diaries were collected at the end of the trial to see whether the use of snack products increased the protein intake and energy intake of the participants. 

The participants received the 3-day food diaries by mail with written instructions on how to fill them in. They were also instructed via phone by the study personnel. The participants returned the food diaries by mail, and the study personnel contacted them by phone if any clarification was needed. The food diaries were analyzed using Fineli, which contains a database of Finnish foods [28].

In addition, a background questionnaire included questions on lifestyle habits, general health, diseases, and the use of medications and dietary supplements. Height, weight, and waist circumference were measured during the baseline visit, and weight and waist circumference were also measured during the end visit. Measurements of weight and height were used to calculate body mass index (BMI kg/m^2^).

### 2.6. Sample Size

The sample size was based on the change in time of the five-time-sit-to-stand test. The reference values used to calculate the sample size are based on the Helsinki Businessmen study (unpublished) [29] and on results obtained from a previous study [22]. Clinically meaningful change in the five-time-sit-to-stand test was estimated to be 3 s. As type 1 error is 5% and power is 80%, we estimated that 44 persons per group were needed. We estimated the dropout percentage to be 15%; therefore, we aimed to recruit 102 persons for the study.

The randomization was performed with computer-generated random numbers, which randomly assigned participants to either the intervention or control group.

### 2.7. Statistical Analysis

Data are presented as means with standard deviation (SD) or as counts with percentages. Statistical comparisons between groups were made using t test for continuous variables and Pearson’s chi-square for categorical variables. Mean changes in primary and secondary outcomes between three months and baseline values were assessed using analysis of covariance (ANCOVA) with baseline values as covariates. Effect size (d) was calculated by using the method of Cohen where an effect size of 0.20 is considered small, 0.50 moderate, and 0.80 large. CIs for the effect sizes were obtained by bias-corrected bootstrapping (10,000 replications). Statistical analyses were performed using statistical software (Stata, release 17.0, Stata Corp., College Station, TX, USA).

## 3. Results

In total, we recruited 101 instead of the calculated 102 participants of which 94 finalized the trial (44 participants in the intervention group, 50 in the control group). The flowchart of the study is presented in Figure 1. The intervention and the control groups were similar in most of the baseline characteristics. SARC-F score was 2.2 and 2.3 points in the control and intervention groups, respectively. The baseline characteristics of the participants are presented in Table 3.

The main outcome of the study, the change in the five-time-sit-to-stand test, did not differ between the intervention and control groups at the end of the trial (intervention −2.3 (95% CI −3.9 to −1.6) vs. control −2.2 (95% CI −3.2 to −1.2), *p* = 0.29). Thus, both groups somewhat improved in the five-time-sit-to-stand test. Of the secondary outcomes, the change in total SPPB score differed significantly (+0.8 points (95% CI +0.5 to +1.1) in the intervention group vs. 0.2 points (95% CI −0.2 to +0.7) in the control group), favoring the intervention group (*p* < 0.020) (Table 3). The effect size was moderate. The improvement in the intervention group derives mainly from the improved balance test in the intervention compared with the control.

Protein intake increased significantly in the intervention group due to the intervention compared to the control group; +14 g in the intervention group vs. +2 g in the control group (*p* < 0.001) (Figure 2). There was no significant change in energy intake or in participants’ weights between the baseline and follow-up in either group. Other secondary outcomes (walking speed, hand grip strength, Trail Making Test A and B, health-related quality of life by RAND-36) did not change significantly due to the intervention (Table 4).

The compliance with using the snack products in the intervention group was 92.1%, and self-reported compliance with performing the five-movement exercise routine was 81.8% and 70.7% in the intervention and control groups, respectively (*p* = 0.015).

## 4. Discussion

In our study, supplementation with a snack product containing both MFGM and hydrolyzed protein for 12 weeks did not lead to a significant difference in five-time-sit-to-stand test in the intervention group compared to the control group in older women. However, the physical performance-related balance test and the total SPPB score changed significantly, favoring the intervention group.

Some previous studies have explored the effects of MFGM supplementation combined with exercise on leg muscle mass, muscle fiber velocity, physical performance, and frailty status in older adults and physical agility in middle-aged adults [7,12,13,14,15]. Unlike in these studies on MFGM, our primary outcome (five-time-sit-to-stand test) did not change between the intervention and control groups. One of the key differences between our study and these studies was that our study did not include mandatory exercise. Instead, we provided the participants in both groups with an exercise routine with written instructions that they were advised to perform independently at home without further supervision. The intervention group was also advised to consume the snack product after daily physical activity, which could have potentially enhanced the effect of MFGM and protein. Although the participants self-reported whether they had performed the routine daily, the exercise probably was not as effective as a mandatory, regular, progressive, instructor-based type of exercise. Thus, these results suggest that MFGM and protein supplementation may require additional, more structured exercise to improve muscle strength clinically significantly. The aim of this study was to test whether supplementation alone could improve physical performance, and therefore, both groups were instructed to exercise. In addition, we tested whether our snack product could have had cognitive benefits for our participants, since infant formulas supplemented with MFGM have in some studies enhanced infants’ cognitive development compared to standard infant formula [9,10]. However, we did not observe any cognitive benefits in the intervention group.

Our study further differed from previous studies in respect of the study product; instead of giving the participants MFGM tablets, we offered the intervention participants a snack product high both in MFGM and protein. As the participants had lower than recommended protein intake at baseline, increased protein intake could have had further benefits on physical performance-related outcomes. The protein was also hydrolyzed, which has been shown to enhance absorption and augment postprandial amino acid availability, an important regulator of muscle protein synthesis [17]. A third difference in the study design was that our snack products contained more MFGM than previously reported studies [7,12,13] (3.6 to 3.9 g vs. 1 g). MFGM was deemed safe at 6.5 g/day in a previous study [8], and it was generally well tolerated in our study population.

The balance test score of the SPPB changed significantly, as did the SPPB total score, mainly due to the balance and walking speed scores, favoring the intervention group. In fact, the effect size (ES) for the difference between changes in groups (ES 0.42) was quite good, indicating significant improvement. Similarly, in line with our results, MFGM without mandatory exercise also improved balance measured with a one-leg stand test with eyes closed in community-dwelling Japanese adults (n = 113) aged 50–70 years [30]. Although our results seem similar, our study products were also high in protein; thus, additional protein intake could have also played a role in these findings, especially since the participants had lower than recommended protein intake at baseline. Balance is crucial in preventing falls and injury in older people, and it is necessary for performing daily activities and leading an independent life. Maintaining balance is a complex task that requires coordination of vestibular function, visual function, muscle strength, and the sensory nervous system. MFGM with voluntary exercise has enhanced neuromuscular junction development in middle-aged animals [11] and suppressed the progression of age-related neuromuscular junction degeneration in older animals [31]. Moreover, the effect of MFGM intake with exercise was not restricted to neuromuscular junctions but extended also to the structure and function of peripheral nerves [31]. In particular, the function of peripheral nerves could affect the balance of older people due to age-related changes in these physiological systems, as well as in muscle and bone, which may likely contribute to an increased risk of falls in older people [31]. Thus, balance could be an important target for interventions in older people. Improving balance by adding MFGM and protein-rich snacks to the daily diets of older people could, thus, potentially support healthy and active aging.

Our study has several limitations. The most important limitation arises from the failure to produce a change in the primary outcome. Since our study was designed based on the primary outcome including sample size, the interpretation of secondary outcomes and generalization of the results should be cautiously considered, because the failure of the primary outcome may significantly affect the statistical power of secondary outcomes. Our intention was to find participants who experienced some mobility limitations, and we thus carefully screened the participants with the validated SARC-F questionnaire [18]. However, the mean baseline SPPB total score of the participants was >10 points, which generally indicates robustness. This might cause a ceiling effect in physical performance-related outcomes such as muscle strength and walking speed, therefore making it difficult to observe clinically meaningful changes in these outcomes. Despite this, the ES was quite good, showing a meaningful difference between the groups. It is thus possible that a group with a lower physical performance at baseline could have improved more in response to MFGM and protein supplementation. A further limitation of the study derives from the COVID-19 pandemic. The pandemic challenged and delayed carrying out our study and forced us to make some changes in the original study design by limiting intervention and physical contact with the participants due to the vulnerability of the target population. The pandemic might also have changed the typical physical activity patterns of the participants, thus reducing their daily activity. In our study, we had only dropouts in the intervention group. Since we only had two types of products, a few (n = 2) participants did not like the taste of the products or became bored consuming the same products and consequently dropped out of the study. A few others (n = 2) dropped out due to bowel symptoms, one due to weight gain, one due to allergy, and one due to worsening rheumatic arthritis symptoms. Fortunately, the statistical power of the study remained unaffected by the dropouts. For future studies and for practice, it would be beneficial to have more variety of snack products to match the varied tastes of people. Nevertheless, for those who remained in the study, the self-reported compliance with consuming the study snacks was very high. There was a difference in self-reported compliance with performing the exercise routine between the intervention and control groups; the intervention participants reported higher compliance than the control group (81.8% vs. 70.7%). However, the five-time-sit-to-stand test improved similarly in both groups. One limitation was also that the study was not blinded. Since only the intervention group received a snack product, it was not possible to keep the study blinded.

The strengths of the study derive from the study design. The study was a randomized controlled trial. Both intervention and control participants were advised to exercise; therefore, we could evaluate whether the supplement has an effect on physical performance. The participants were a homogeneous group of Caucasian older women of >70 years of age. As previously reported studies on MFGM and physical performance-related outcomes have mainly been carried out in the Japanese population, our study adds to the scientific knowledge of the effects of MFGM supplementation. We carefully selected people with lower than recommended protein intake at baseline, as well as those who reported having some mobility limitations. All the tests performed (e.g., physical performance, muscle strength) and measurements carried out were validated. The study personnel performing the measurements were experienced in assessing physical performance tests and further received unified training. Our study showed similar results to a previous study by Kokai et al., whose study subjects were somewhat younger, healthy Japanese volunteers. Similar to our study, the balance test was also a secondary outcome [30]. If our results would be corroborated in future studies, adding MFGM with or without protein to the diets of older people could be a feasible way to improve their balance and allow them to lead more active and healthy lives. Benefits in balance could possibly be achieved even without increasing exercise [30]. Carefully considered primary outcomes and well-designed study designs are of the essence in future studies.

In conclusion, our study suggests that the combination of MFGM and protein improved physical performance related to balance and total SPPB score in community-dwelling older women. In the future, balance and fall prevention could be considered as primary outcomes in studies with MFGM or MFGM combined with protein, especially in study designs without mandatory exercise. It would also be interesting to see the results of RCTs on diverse groups of older people to learn who would possibly benefit the most from these types of interventions.

## Figures and Tables

**Figure 1 nutrients-15-02922-f001:**
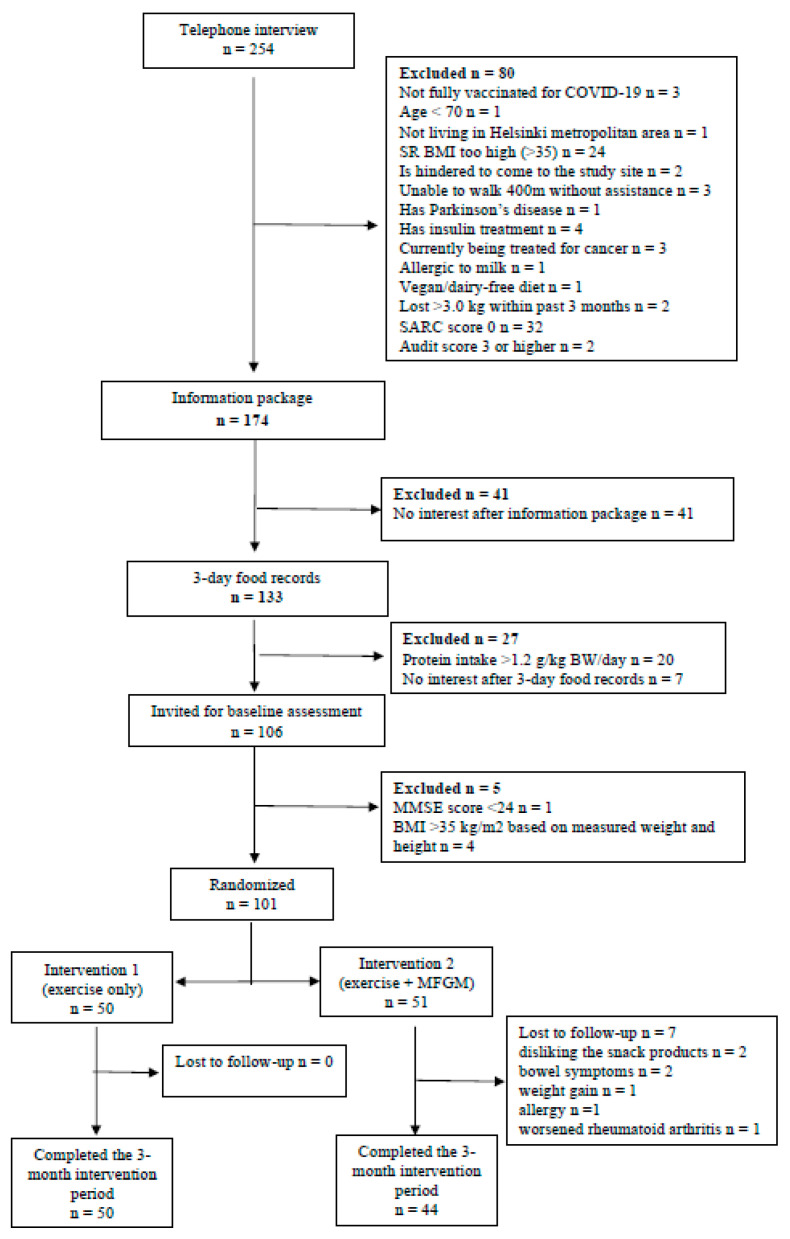
Flowchart of the study.

**Figure 2 nutrients-15-02922-f002:**
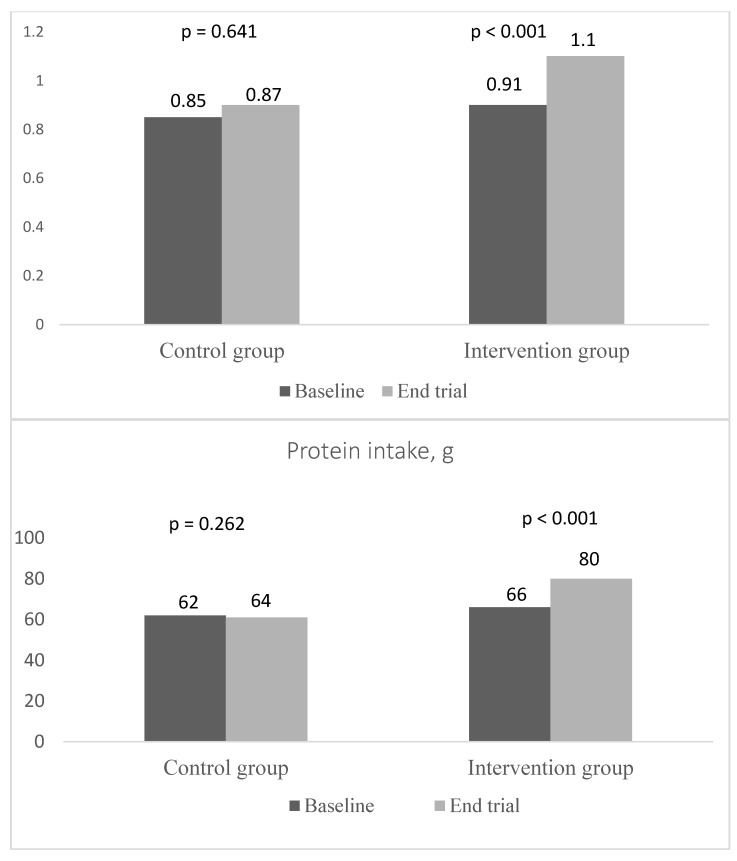
Protein intake total g and g/kg body weight (BW)/d in the control and intervention groups at baseline and at the end of the trial.

**Table 1 nutrients-15-02922-t001:** Inclusion and exclusion criteria.

**Inclusion Criteria:**
Home-dwelling female ≥ 70 years of age
SARC-F > 0
Protein intake < 1.2 g/kg body weight/d according to a 3-day food diary
Subjective walking ability 400 m
**Exclusion Criteria:**
Poor cognition or neurocognitive disorder, MMSE < 24
SARC = 0
AUDIT ≥ 3
Bed- or wheelchair-bound
Self-reported severe kidney malfunction
Acute cancer treatment less than a year ago, except for basal cell carcinoma
Parkinson’s disease
Insulin-dependent diabetes
Special diet that prevents use of the snack product provided by the study
Spends long time away from the metropolitan area during the study
Currently participates in another lifestyle intervention
Unintentional weight loss > 3 kg in 3 months
Severe obesity (BMI over 35 kg/m^2^)
Evaluation by the researchers that the subject will not be able to finish the study

**Table 2 nutrients-15-02922-t002:** Nutritional contents of the research snack products.

Snack Product	Protein Powder	Milkshake
Nutritional Content	Serving Size 30 g	Serving Size 250 mL
Energy, kcal	140	190
Protein, g	23	23
Fat, g	4	4
Carbohydrates, g	2	16
Lactose, g	0	0
MFGM, g	3.9	3.6
Phospholipids, g	1.3	1.2

**Table 3 nutrients-15-02922-t003:** Baseline characteristics of the participants in the control and intervention groups.

Group	Control	Intervention
Characteristics	N = 50	N = 44
Age, mean (SD)	76 (4)	75 (4)
Education, n (%)		
elementary school or less	2 (4)	3 (6)
vocational school	8 (16)	7 (14)
upper comprehensive school	7 (14)	6 (12)
upper secondary school	7 (14)	11 (22)
institute of higher education	25 (51)	23 (46)
Financial status, n (%)		
good	23 (48)	18 (36)
moderate	24 (50)	30 (60)
poor	1 (2)	2 (4)
Current smoker, n (%)	2 (4)	3 (6)
AUDIT, mean (SD)	1.7 (1.1)	2.1 (1.6)
SARC-F score, mean (SD)	2.3 (0.9)	2.2 (1.2)
MMSE score mean (SD)	28.8 (1.6)	28.6 (1.7)
Weight, kg, mean (SD)	75 (11)	74 (11)
BMI, kg/m^2^, mean (SD)	28.3 (3.6)	28.3 (3.6)
Waist circumference, cm, mean (SD)	95.6 (11.1)	93.0 (10.1)
Energy, kcal, mean (SD)	1559 (275)	1644 (388)
Protein intake g, mean (SD)	62 (13)	66 (15)
protein g/kg BW/day	0.85 (0.20)	0.90 (0.18)
Protein intake < 1.0 g/kg BW/day, n (%)	15 (30)	16 (30)
Diseases during the last 12 months, n (%)		
Diabetes	4 (8)	7 (14)
Coronary artery disease	4 (8)	4 (8)
Cardiac failure	2 (4)	1 (2)
Arterial fibrillation	4 (8)	5 (10)
Other cardiac disease	3 (6)	2 (4)
Hypertension	31 (63)	25 (51)
Other transient ischemic attack	4 (8)	4 (8)
Foot circulatory disorder	10 (20)	5 (10)
Asthma	5 (10)	8 (16)
Osteoarthritis	31 (63)	34 (69)
Thyroid disease	15 (30)	10 (20)
Cancer	16 (32)	11 (22)
Number of medications, n (%)	4.6 (3.3)	3.9 (2.7)

AUDIT = Alcohol Use Disorders Identification Test; SARC-F = sarcopenia–frailty questionnaire; MMSE = Mini-Mental State Examination: BMI = body mass index; BW = body weight.

**Table 4 nutrients-15-02922-t004:** Primary and secondary physical performance, cognition, and health-related quality of life outcomes of the study at baseline and the change in 12 weeks.

	Baseline		Change from Baseline to 12 Weeks		*p*-Value *	Effect Size (95% CI) **
Primary and Secondary Outcomes	Control N = 50 Mean (SD)	Intervention N = 44 Mean (SD)	Control Mean (95% CI)	Intervention Mean (95% CI)		
5-time-chair-stand test, s	14.9 (4.2)	13.8 (3.4)	−2.2 (−3.2 to −1.2)	−2.3 (−3.0 to −1.6)	0.29	0.05 (−0.36 to 0.47)
Balance test, s	3.70 (0.58)	3.82 (0.45)	−0.06 (−0.21 to 0.10)	0.09 (−0.05 to 0.23)	0.022	0.29 (−0.06 to 0.69)
Walking speed, m/s	1.27 (0.37)	1.30 (0.33)	0.03 (−0.03 to 0.09)	0.06 (−0.01 to 0.13)	0.36	0.15 (−0.25 to 0.55)
Total SPPB score	10.0 (1.5)	10.3 (1.5)	0.2 (−0.2 to 0.7)	0.8 (0.5 to 1.1)	0.020	0.42 (0.04 to 0.74)
Hand grip strength, kg	20.4 (5.1)	21.4 (5.4)	0.9 (0.1 to 1.7)	1.0 (0.0 to 1.9)	0.51	0.03 (−0.38 to 0.43)
Trail Making Test, s						
A	42.1(15.2)	47.8 (17.9)	−5.7 (−9.1 to −2.3)	−6.7 (−10.7 to −2.7)	0.50	0.08 (−0.34 to 0.54)
B	108.2 (41.1)	112.9 (49.2)	−14.1 (−23.4 to −4.8)	−8.0 (−20.7 to 4.6)	0.22	−0.16 (−0.60 to 0.26)
RAND-36						
Physical Health Summary	41.7(9.0)	42.2 (8.6)	1.8 (−0.2 to 3.9)	1.3 (−0.7 to 3.4)	0.80	−0.07 (−0.51 to 0.34)
Mental Health Summary	51.8(9.7)	50.9 (11.1)	−1.2 (−3.5 to 1.1)	−1.5 (−4.1 to 1.1)	0.75	−0.04 (−0.45 to 0.37)

* Baseline adjusted; ** 95% confidence intervals (CIs) were obtained by means of bias-corrected bootstrapping (10,000 replications). − favoring control, favoring intervention.

## Data Availability

The data are not publicly available due to the signed consent agreements around data sharing, which only allow access for the researchers of the study following the project purposes. Requestors wishing to access the data used in this study can make a request to M.H.S. The request will be subject to approval and formal agreements regarding confidentiality and secure data storage being signed.

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
