# Peer review of "Effect of Milk Fat Globule Membrane- and Protein-Containing Snack Product on Physical Performance of Older Women—A Randomized Controlled Trial"

_nutrients, 2023, doi:10.3390/nu15132922_

Round 1

Reviewer 1 Report

This study investigated the effect of protein intake and milk fat globule membrane (MFGM) on sarcopenia. It is interesting to know that the combination of MFGM and protein may improve physical performance related balance of older women. I have some comments as shown below.

1.     What is the full name of SPPB score in the abstract?

2.     Why the author investigate MFGM protein not whey protein? As far as I know, whey protein has also been reported to have improvement on sarcopenia of elderly people?  Why the author include a group receiving the same amount of proteins with whey proteins as control?

3.     I would like to know the composition of MFGM. Was it a whey protein ingredient with MFGM proteins or purified MFGM proteins?

4.     Except for the chocolate milkshake (a serving of 250 ml) and protein powder (a serving of 30g), what did the participants eat during the day? Will the diet be different in total protein consumption? How did the author to keep the daily protein intake the same in two groups?

5.     Why the authors focused on elderly women instead of men?

6.     Although, we see that the balance performance of intervention group was improved after consumption of protein hydrolysate and MFGM protein. The effect could be related to both protein hydrolysate and MFGM protein.

7.     Could the author add discussion about why the protein and MFGM could improve the balance of elderly women?

Author Response

Thank you for your positive response of our manuscript,  titled “Effect of milk fat globule membrane and protein containing snack product on physical performance of older women - a randomized controlled trial”. We appreciated the constructive criticism and comments and our responses are presented point-by-point.

Our point-by-point response is attached below.

Best,

in behalf of all authors

Satu Jyväkorpi

Reviewer 2 Report

Comments to Author

This article examined whether snacks containing milk fat and protein improve physical performance in older women in an RCT.

It's a very interesting paper, but it needs minor revision.

Major Comments

1. Self-report compliance appears to be fairly high in both the control and intervention groups. How was the compliance of the subjects compared to previous studies? Nevertheless, it is better to show in the discussion what you think is the cause of the lack of difference in the five sit-to-stand tests.

2. The exclusion criteria include severe kidney disease, however, it is better to show how you handled a disease that requires dialysis or other protein restriction.

3. Please indicate your reasons for targeting those over 70 years old.

Minor Comments

1. Table1 SARC → SARC-F

2. Line145 

It’s better to include citations “using a standard protocol”.

Author Response

Thank you for your positive response of our manuscript, originally titled “Effect of milk fat globule membrane and protein containing snack product on physical performance of older women - a randomized controlled trial”. We appreciated the constructive criticism and comments of the reviewers and editors and our responses are presented point-by-point.

Please find attached our point-by-point response for the reviewer 2.

Sincerelly,

In behalf of all the authors,

Satu Jyväkorpi

Round 2

Reviewer 1 Report

The authors have addressed my comments.